

# Weaknesses in dust emission modelling hidden by tuning to dust in the atmosphere

Adrian Chappell[1], Nicholas P. Webb[2], Mark Hennen[1], Charles Zender[3], Philippe Ciais[4,5], Kerstin Schepanski[6], Brandon L. Edwards[2], Nancy Ziegler[7], Sandra Jones[7], Yves Balkanski[4], Daniel Tong[8], John F. Leys[9,10], Stephan Heidenreich[9], Robert Hynes[9], David Fuchs[9], Zhenzhong Zeng[11], Marie Ekström[1], Matthew Baddock[12], Jeffrey Lee[13], Tarek Kandakji[14].

[1]School of Earth and Environmental Sciences, Cardiff University, Cardiff CF10 3XQ, UK.
[2]USDA-ARS Jornada Experimental Range, Las Cruces, NM 88003, USA.
[3]Department of Earth System Science, University of California, Irvine, USA.
[4]Laboratoire des Sciences du Climat et de l'Environnement, CEA CNRS UPSACLAY, Gif-sur-Yvette, France.
[5]Climate and Atmosphere Research Center (CARE-C), The Cyprus Institute, 20 Konstantinou Kavafi Street, 2121 Nicosia, Cyprus.
[6]Institute of Meteorology, Freie Universität Berlin, Germany.
[7]US Army Engineer Research and Development Center, Cold Regions Research and Engineering Laboratory (CRREL), 72 Lyme Rd, Hanover, NH 03755-1290, USA.
[8]Department of Atmospheric, Oceanic and Earth Sciences, George Mason University, Fairfax, VA 22030 USA.
[9]Department of Planning, Industry and Environment, NSW, Australia.
[10]The Fenner School of Environment and Society, Australian National University, Australia.
[11]School of Environmental Science and Engineering, South University of Science and Technology of China, Shenzhen 518055, China.
[12]Geography and Environment, Loughborough University, Loughborough, UK
[13]Department of Geosciences, Texas Tech University, Lubbock, TX 79409, USA.
[14]Yale Center for Earth Observation, Yale University, New Haven, CT 06520, USA

Correspondence to: Adrian Chappell (chappella2@cardiff.ac.uk).

**Abstract.** Dust emissions influence global climate while simultaneously reducing the productive potential and resilience of landscapes to climate stressors, together impacting food security and human health. Vegetation is a major control on dust emission because it extracts momentum from the wind and shelters the soil surface, protecting dry and loose material from erosion by winds. Many of the traditional dust emission models (TEM) assume that the Earth's land surface is devoid of vegetation, then adjust the dust emission using a vegetation cover complement, and finally calibrate the magnitude of simulated emissions to dust in the atmosphere. We compare this approach with a novel albedo-based dust emission model (AEM) which calibrates Earth's land surface shadow (1-albedo) to shelter depending on wind speed, to represent aerodynamic roughness spatio-temporal variation. We also compared the TEM and AEM dust emissions with estimates of dust in the atmosphere using dust optical depth frequency (DOD) and satellite observed dust emission from point sources (DPS). We show that during the same period, the DOD frequency exceeds by two orders of magnitude DPS frequency (RMSE_DOD=151 days). Also relative to DPS frequency, both models over-estimated dust emission frequency but by only one order of magnitude (RMSE_TEM=27 days; RMSE_AEM=20 days) and showed strong relations with DPS frequency, suitable for





calibrating models to observed dust emission. Theoretically, the TEMs are incomplete in their formulation, which despite the pragmatic adjustment using the vegetation cover complement, causes dust emission to be highly dependent on wind speed and over-estimates large ($>0.1$ kg m$^{-2}$ a$^{-1}$) dust emission over vast vegetated areas. Consequently, the TEMs produce considerable false change in dust emission, relative to the AEM. Since the main difference between the dust emission models is the treatment of aerodynamic roughness, our results suggest that its crude representation in the TEMs has caused large, previously unknown, uncertainty in Earth System Models (ESMs). It is difficult to avoid our conclusion, also raised by others, that tuning dust emission models to dust in the atmosphere has hidden for more than two decades, these TEM modelling weaknesses and its poor performance. The AEM overcomes these weaknesses and improves performance before calibration. The major advantage for ESMs, is that the AEM can be driven by intrinsic prognostic albedo to represent the fidelity of drag partition physics and reduce uncertainty of aerosol effects on, and responses to, contemporary and future environmental change.

## 1 Introduction

Vegetation attenuates dust emission by extracting momentum from the wind and sheltering a portion of the downstream soil. By reducing wind speeds ($U$) at the soil surface, vegetation makes it more difficult to overcome the entrainment threshold for initiation of streamwise sediment flux (hereafter entrainment threshold) and consequent emission of dust particles by saltation bombardment and abrasion. Notably, the influence of vegetation sheltering is wind speed dependent (aerodynamic roughness) and both aerodynamic drag and partitioning of wind friction velocity between roughness elements and the soil, respond nonlinearly to changes in wind speed. Calculation of the stream-wise sediment flux density $Q$ (g m$^{-1}$ s$^{-1}$) on a smooth soil for a given particle size fraction ($d$) on the particle size distribution ($i$) requires the total wind friction velocity $u_*$ (m s$^{-1}$), created by all scales of roughness at the Earth's surface, the air density $\rho_a$ (g m$^{-3}$), the acceleration due to gravity $g$ (m s$^{-2}$), a dimensionless fitting parameter $C$ and the bare, smooth (no roughness elements) entrainment threshold of sediment flux $u_{*ts}(d)$ (m s$^{-1}$) (Kawamura, 1951). It is now commonly rewritten in the dust modelling literature with the typographic correction and reformulated ratios (White, 1979) which require a cubic term:

$$Q(d) = C \frac{\rho_a}{g} u_*^3 \left(1 - \frac{u_{*ts}^2(d)}{u_*^2}\right)\left(1 + \frac{u_{*ts}(d)}{u_*}\right)\begin{cases} u_* > u_{*ts}(d) \\ 0 \end{cases}. \tag{Eq. 1}$$

In ESMs or reanalysis models over large areas (large pixels), with horizontal resolutions that are typically on the order of 50 km, modelled wind speed at 10 m ($U_{10}$) is used to calculate the available above canopy $u_*$. In recognition that vegetation exerts drag on the wind, $u_*$ must then be partitioned between the roughness elements (typically vegetation), and that available for driving flux at the soil ($u_{s*}$). The $u_{*ts}$ is adjusted by a soil moisture function $H(w)$ (Fécan et al., 1998) and $R = \frac{u_{s*}}{u_*}$ (Raupach et al., 1993) the wind friction velocity ratio representing the roughness-induced drag partition (Marshall,





1971). The $u_{s*}$ is required for sediment flux equations where $u_{s*} \neq u_*$ and $Q$ (Eq. 1) is modified (Darmenova et al., 2009) in
the TEMs:

$$Q_{TEM} = C \frac{\rho_a}{g} u_*^3 \left(1 - \frac{(u_{*ts}H(w)/R)^2}{u_*^2}\right)\left(1 + \frac{u_{*ts}H(w)/R}{u_*}\right) \begin{cases} u_* > u_{*ts}H(w)/R \\ 0 \end{cases}.$$ (Eq. 2)

Instead of estimating directly $u_{s*}$, the $u_{*ts}$ is divided by $R$ for the model implementation to account for the drag partition and
to make use of $u_*$ (Webb et al., 2020). Following this approach, this form (Eq. 2) is incomplete because $u_*$ (on the left-hand
side, the magnitude calculation) must be multiplied by $R$ before it is cubed (Webb et al., 2020). The entrainment threshold
($u_{*ts}$) is calculated at the grain scale as a function of grain diameter, density and inter-particle cohesion (Shao et al., 1996).
However, the above canopy $u_*$ is for an area, which requires $u_{*ts}$ to be represented over the same area, which it is not.

The substantive issues for dust emission modelling are that the incomplete form of $Q_{TEM}$ (Eq. 2) has been widely
adopted in TEMs in which large area estimates of wind speed are typically used, the correct values of $R$ are not known (for
every pixel and every time step) and $u_{*ts}$ is not scaled correctly. The common approach to modelling dust emission in ESMs
uses globally constant values of aerodynamic roughness length ($z_0$), which are static over time and fixes $R(z_0) \approx 0.91$. The
values of $z_0$ are 'pre-tuned' to the Earth's bare (devoid of vegetation) land surface, and therefore tend to maximize dust
emission. This emission is then reduced by a function of vegetation cover and ultimately 'tuned' down to match observed
dust in the atmosphere. In practice, models define geographically some preferential dust sources (Ginoux et al., 2001; Tegen
et al., 2002; Zender et al., 2003a; Mahowald et al., 2010; Woodward, 2001; Evans et al., 2016). A second, more recent
approach uses satellite remote sensing to provide spatially heterogeneous estimates of $z_0$ only for arid and semi-arid regions,
but fixed over time (Greeley et al., 1997; Roujean et al., 1997; Marticorena et al., 2004; Prigent et al., 2012; Prigent et al.,
2005). With this second approach it is still challenging to estimate $R$. Here we focus on the impact for large scale TEMs
where $R(z_0)$ is fixed over space and time with the incomplete formulation for $Q_{TEM}$ (Eq. 2).

In our new formulation called AEM for Albedo-based dust Emission Model, the spatio-temporal variation in $u_{s*}$ is
simulated using the concept that aerodynamic sheltering of vegetation is proportional to its shadow (1-albedo) (Chappell et
al., 2010; Chappell and Webb, 2016). This albedo-based approximation of the drag partition was investigated and tested to
provide an area-weighted value, shown to be scale invariant (Chappell et al., 2018; Chappell et al., 2019; Ziegler et al.,
2020). This approach enables direct calculation of $u_{s*}$ given measurements of albedo from satellites, and the correct
formulation for sediment flux and dust emission

$$Q_{AEM} = C \frac{\rho_a}{g} u_{s*}^3 \left(1 - \frac{(u_{*ts}(d)H(w))^2}{u_{s*}^2}\right)\left(1 + \frac{u_{*ts}(d)H(w)}{u_{s*}}\right) \begin{cases} u_{s*} > u_{*ts}(d)H(w) \\ 0 \end{cases}.$$ (Eq. 3)





Notably, this approach retains the long-established entrainment threshold $u_{*ts}$ which at the grain-scale is inconsistent with the new area-weighted albedo-based approach. The threshold value is very likely much smaller than the necessary (but currently unknown) value for entrainment threshold for a 500 m pixel. Consequently, modelled dust emission is expected to be over-estimated. However, this component of the modelling is beyond the scope of this manuscript. The $u_{s*}$ is obtained directly from $\omega_{ns}$, the normalised and rescaled shadow (1-albedo), enabling an albedo-based dust emission model (AEM; see

Appendix for full description of the implementation)

$$\frac{u_{s*}}{U_{10}} = 0.0311 \left( exp \frac{-\omega_{ns}^{1.131}}{0.016} \right) + 0.007.$$  (Eq. 4)

The vertical dust mass flux ($F$; g m$^{-2}$ s$^{-1}$) may be calculated from $Q$ using physically-based schemes (Kok et al., 2014; Shao

et al., 1996). More commonly in regional and global applications and here for the TEM and AEM, $F$ is calculated as an empirical function of $Q$ (Marticorena and Bergametti, 1995):

$$F = EM(d)Q(d)10^{(0.134clay_\% - 6.0)}.$$  (Eq. 5)

The dust emission parameterisation considers the emission flux to be driven by saltation bombardment, with the intensity proportional to $Q$, and the soil's clay content (clay$_\%$ typically <2 µm fraction of soil particles at the soil). We fixed the mass fraction of clay particles in the parent soil to $clay_\%$=20 consistent with previous work (Zender et al., 2003a). The proportion of emitted dust in the atmosphere $M$ for a given particle size fraction ($d$) is dependent on the particle size distribution. We calculated the relative particle size surface area (Marticorena and Bergametti, 1995) ($M$). The vegetation cover function $E$

was originally defined (Marticorena and Bergametti, 1995) as the ratio of bare exposed surface area to total surface area when viewed from directly above (at nadir). It is used to adjust linearly the amount of dust emission by the bare soil fraction. However, sheltering is nonlinear since it depends on the mutual sheltering of the roughness (typically vegetation) structure, configuration and wind speed (Chappell et al., 2010). Theoretically, $R$ in the equations above already accounts for the soil area which is exposed to wind friction velocity relative to that sheltered by upwind roughness elements. Therefore, $E$ is

theoretically redundant in the TEM (Webb et al., 2020). Nevertheless, its use assume $E$=1-$A_v$ where $A_v$ is the area covered by roughness elements, typically vegetation. This $E$ is used in some ESMs so that leaf area index (LAI) or satellite 'greenness' observations e.g., normalized difference vegetation indices (NDVI) can be used as a surrogate of the land surface fraction occupied by green vegetation (Evans et al., 2016; Galloza et al., 2018; Zender et al., 2003a; Sellar et al., 2019). After the sediment flux is calculated, only then is $E$ used to adjust dust emission using the area covered by green vegetation. In

addition, $E$ does not represent 'brown roughness' caused by dormant or dead vegetation or non-erodible stone covered surfaces in dryland regions where most sediment flux and dust emission occurs. This crude model representation of process is a prime example of the influence exerted by the emphasis of parsimony in model implementation. When the TEMs are





applied in dust-climate ESMs it is assumed that this parameterization is adequate for climate projections. In contrast, the albedo-based scheme for sediment flux and dust emission (AEM; Eqs. 3, 4 & 5) represents the drag partition physics without

pre-tuning to a fixed land surface condition, without the need for $E$, and thereby removes these additional sources of uncertainty.

## 2 Methods and Data

### 1.1 Modelled dust emission evaluated against dust emission point sources and dust optical depth

Commonly, aerosol optical depth (AOD) from point (ground-based) or large area Earth observation (EO) data are used to

evaluate the performance and / or calibrate dust emission model simulations (Meng et al., 2021). This approach assumes that: i) dust in the atmosphere represents the dust emission process, and ii) the spatial variation in magnitude and frequency of dust emission in the model is correct. However, we know *a priori* that dust in the atmosphere is only partially related to dust emission because dust concentration is controlled by dust emission magnitude and frequency which varies over space and time, by residence time of dust near the surface which itself is dependent on wind speed, and on dust deposition in the

dust source region, a size dependent process. To understand the extent to which AOD estimates the spatial variation in dust emission magnitude and frequency we calculated the probability of dust occurrence modelled by the dust optical depth (DOD>0.2) using the criteria established previously (Ginoux et al., 2012). We note the stated limitations of DOD to be largely restricted to bright land surfaces in the visible wavebands which implies reduced performance over areas where vegetation is present. To calculate DOD, we used wavebands available from monthly Moderate Resolution Imaging

Spectroradiometer (MODIS; MOD08 M3 V6.1) at a 1-degree pixel resolution (Platnick, 2015). The DOD was retrieved from those pixels in which dust emission was observed from point sources (DPS) in space and time throughout 2001-2016. All available MODIS DOD data were used, quality flags were not used to filter these data.

We described in the previous section how simplifying assumptions are made in TEMs about the dynamics of vegetation sheltering. We also provided a theoretical basis for TEMs formulation to be incorrect. The correct magnitude and

frequency of dust emission per unit area depends on the correct probability that sediment flux occurs, causing dust emission which itself depends on the correct $u_{*ts}H(w)$ (and the correct $R$ in the TEM). However, most dust emission schemes using $u_{*ts}$ assume that the soil is smooth and covered with an infinite supply of loose erodible material which when mobilised causes dust emission in proportion to the clay content. This (energy limited) assumption is rarely justified in dust source regions where (i) the soil is rough due to soil aggregates, rocks or gravels, (ii) sealed with biogeochemical crusts, or (iii)

loose sediment is simply unavailable (Galloza et al., 2018). Here we circumvent these assumptions to improve the constraints on the dust emission modelling evaluation.

We define a satellite observed dust emission point source (DPS) and its probability of occurrence P(DPS>0) as a first order approximation of the probability of sediment flux P($Q$>0) leading to the proportion of dust ($F$) emission P($F$>0) at those locations. The DPS data are from several previous studies in North America (Kandakji et al., 2020; Lee et al., 2012;





Baddock et al., 2011) which identified the locations of dust emissions in New Mexico and Texas between 2001-2016, 2001-2009 and in 2001-2009 in the Chihuahuan Desert and New Mexico using MODIS data at 250 m spatial resolution with visible to thermal infrared wavebands (0.4–14.4mm; **Figure 1a**). Modelled (AEM and TEM) and observed frequencies are aggregated by a 1°x1° grid matrix, normalizing the results to the lowest resolution data (MODIS DOD) (**Figure 1**). For each grid box location, the observed frequency is calculated as the number of DPS observations per year during observation

period (2001 – 2016). The AEM and TEM modelled dust emission frequency describes (F>0) at DPS locations in each grid cell per year during the same period. DOD modelled frequency describes DOD > 0.2 in each grid pixel per year for the same period.

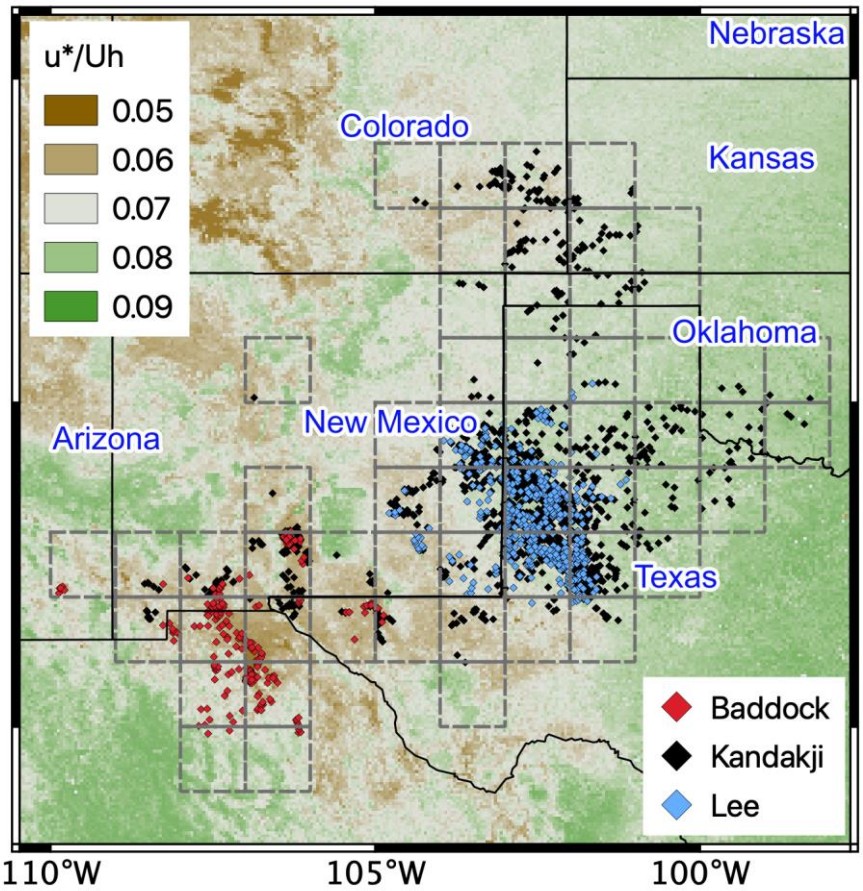

**Figure 1**. Location and publication source (Kandakji et al., 2020; Lee et al., 2012; Baddock et al., 2011) inventory in New
Mexico and Texas between 2001-2016 (Kandakji), 2001-2009 (Lee) and in 2001-2009 in the Chihuahuan Desert and New Mexico (Baddock) using satellite observed dust emission point sources (DPS) set against a background of total wind friction velocity ($u_*/U_{10}$) derived from MODIS albedo (500 m).





At the locations and across the study durations of those DPS data we calculated the AEM and TEM dust emission.
We compared the model estimates during DPS observed occurrence with modelled dust emission determined by the TEM
and AEM. Similarly, during those same DPS observed occurrence we compared the model estimates of dust in the
atmosphere approximated using DOD. For all of those model estimates of dust frequency (DOD, TEM, and AEM),
separately we fitted log-linear regression models which produced regression model parameter coefficients, $R^2$ correlation and
the square root of the sum of squared difference (SSE) between DPS and model predictions to form the RMSE=$\sqrt{\text{SSE}}$/(*N-df)*
where *N* number of data are adjusted by the degrees of freedom (*df*=number of independent dust emission model
parameters).

## 1.2 Large scale dust emission modelling, mapping spatial variation and change detection

We used contemporary (2001-2020) Earth observation data including spatially and temporally varying wind speeds (at 10
m), soil moisture (0-7 cm) and soil temperature (to represent frozen ground which inhibits sediment flux) from the latest
ERA5-Land (Muñoz Sabater, 2019) (hourly; ~11 km). The use of these data does not imply priority over any other data. We
recognize that there are different qualities to different model data as evident in their wind fields (Fan et al., 2021). Where
applicable, we used the same data in both TEM and AEM to consider the relative differences. We used the TEM (**Eqs. 1 &
5**) with $R(z_0, z_{0s})\approx 0.91$ fixed over space and static over time. Following the current practice, we calculated $u_*$ from the
modeled 10 m wind velocity using the logarithmic layer profile theory and aeolian roughness length (Darmenova et al.,
2009) (details are provided in the Appendix). In the TEM we allowed soil moisture to vary and used MODIS Normalised
Difference Vegetation Index (NDVI only in the TEM) data to calculate the bare soil fraction *E*. For comparison, we used the
AEM (**Eqs. 3, 4 & 5**) with soil wind friction velocity $u_{s*}/U_{10}$ obtained from MODIS albedo (MCD43A3; Collection 6)
varying daily, every 500 m pixel across the study area. MODIS is aboard polar-orbiting satellites which cause incomplete
coverage. However, the variation in roughness at the daily scale is so small that we were able to smooth the available data to
improve the coverage. Soil clay content was represented with a digital soil texture map (Dai et al., 2019) and used in both
models (see Methods).

All data were available from the catalogue of the Google Earth Engine (GEE) (Gorelick et al., 2017) which then
required no downloading and reformatting. We used the Java script coding environment to calculate daily dust emission (kg
m$^{-2}$ y$^{-1}$). Given the availability of DPS validation data at sites in south-western USA, we restricted the mapping to North
America including dust source regions bordering the USA. Testing the code and visualising the results for smaller time
periods across the study area was almost instantaneous in the GEE. Data processing at 500 m and daily resolution between
2001-2020 across North America took typically less than 12 hours. These data were exported from the GEE for the
calibration / validation in a Python coding environment and images (TIF) from the GEE were also exported for manipulation
and presentation using ArcGIS Pro.

At the sites and days when dust was observed using dust emission point sources (DPS) we compared it with the dust
emission produced by TEM, AEM and dust in the atmosphere using DOD. For the year 2020 and the main dust emission





months of March-May (MAM), we analysed across North America the spatial variation of the main controlling variables (wind and aerodynamic roughness) and dust emission produced by TEM and AEM. The dust emission of both models is restricted to wind speeds between 8.5-9.5 m s⁻¹ to emphasise the difference in the modelling approaches, which would
otherwise be hidden by taking the average for all wind speeds. Finally, we also map the difference in driving variables during MAM for the year 2001 compared with the year 2020. The dust emission on dust days is similarly compared to obtain the mean difference. That mean difference is then tested for significance using the minimum detectable change (MDC) framework (Woodward, 1992; Webb et al., 2019) and the results are displayed. The minimum detectable change (MDC) was established using critical values for false acceptance and false rejection ($\alpha = 0.05$; $\beta = 0.05$, respectively) and the change
in dust emission which did not exceed the MDC, was set to 0 (not detectable=white). Details of how the MDC was calculated are described in the Appendix.

## 3 Results

### 3.1 The impact of incorrect formulation and fixed drag partition (R) on dust emission modelling

We simulated dust emission separately for a smooth and rough surface with wind speed varying between 0-12.5 m s⁻¹
(**Figure 2a**). The TEM is shown with a fixed aerodynamic roughness length for the landscape $z_0$=100 µm and the soil $z_{0s} =$ 33.3 µm following previous studies (Zender et al., 2003a), which fixes $R(z_0) \approx 0.91$ and assumes that the Earth's land surface is devoid of vegetation roughness and static over time. With $E$=1, dust emission is unadjusted and increases along the upper (dashed) curve as wind speed increases. When the land surface is partially covered in vegetation it becomes rough and $E$=0.5, all other conditions remaining the same. In this case, dust emission increases as wind speed increases but at a
consistently reduced rate (the lower dashed curve for the rough situation). The implication is that the same amount of dust emission is produced for a range of wind speeds (e.g., 8-9.2 m s⁻¹) regardless of whether the land surface is smooth or rough (open square to filled square).

In contrast, the albedo-based dust emission model (AEM) for the smooth situation ($u_{s*}/U_{10}$=0.035; dotted line) produces larger dust emission than the TEM for the same 8 m s⁻¹ wind speed (**Figure 2a**). In a rough situation ($u_{s*}/U_{10}$=0.022) dust emission declines along the same curve to almost zero. Despite a larger wind speed of 9.2 m s⁻¹, the rough surface causes the surface wind friction velocity to decrease, barely exceeding the entrainment threshold, and dust emission to be considerably reduced. The implication is that the increase in roughness is sufficient to overcome the increase in wind speed and causes dust emission to be much smaller. The interplay between wind speed and roughness influences surface wind friction velocity which is essential to accurate and precise dust emission estimates.



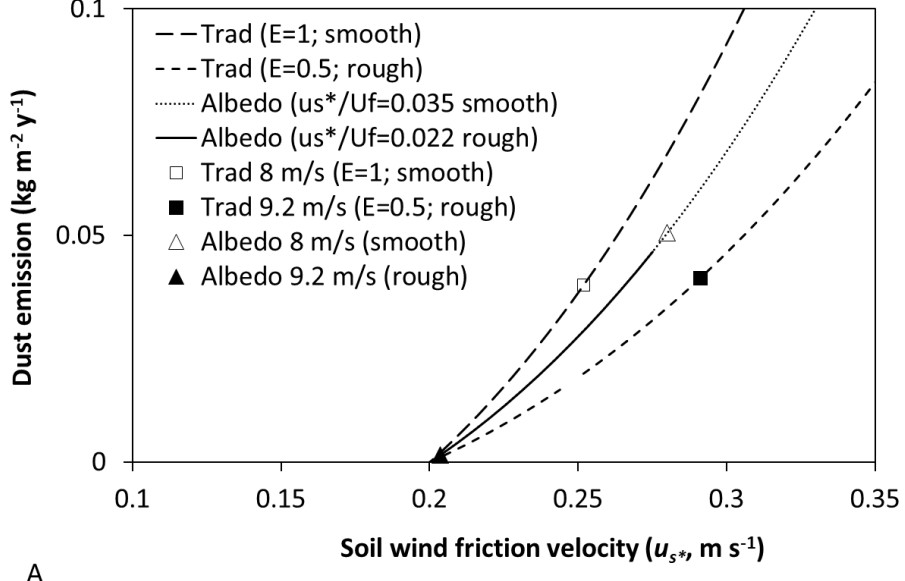


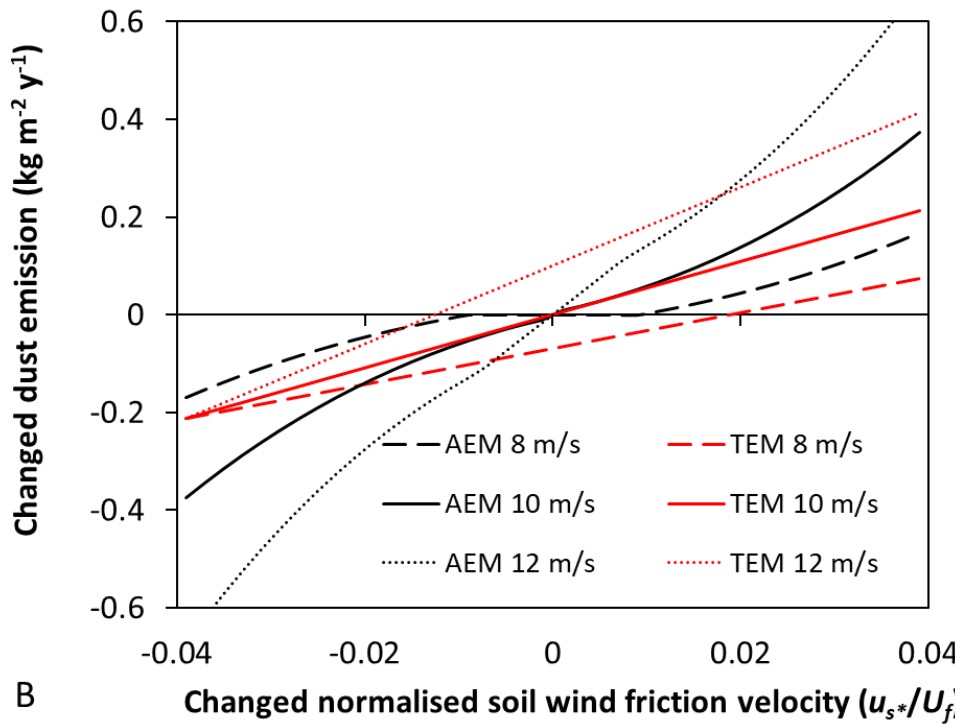

**Figure 2.** Dust emission (kg m$^{-2}$ y$^{-1}$) simulations shown with varying soil wind friction velocity (**A**) and with varying soil wind friction velocity (**B**) normalised by wind speed at 10 m height ($U_{10}$) using fixed entrainment threshold $u_{*ts}$=0.2 m s$^{-1}$,





clay=10%, soil moisture function $H(w)$=1 and the bare soil function $E$ depending on the roughness. The TEM was
implemented (Eqs. 2 & 5) with fixed aerodynamic roughness length ($z_0$) and consequently fixed $R(z_0)\approx0.91$. The albedo-
based dust emission was implemented (Eq. 3, 4 & 5) as described in the main text with details in the Appendix.

These findings are expected based on the theory described above in the Methods section: the TEM is driven by
wind speed attenuated by aerodynamic roughness which is fixed over space and static over time, and dust emission is
subsequently reduced by a bare soil fraction ($E$ based on vegetation cover). Consequently, wherever and whenever wind
speed exceeds the entrainment threshold, the TEM will produce sediment flux and dust emission. To illustrate this point,
**Figure 2b** shows change in dust emission with change in $u_{s*}$ normalized by wind speed $U_{10}$. In other words, **Figure 2b**
shows how dust emission changes as roughness changes in either space and / or time for the TEM and AEM. Since the
influence of wind speed is removed on the x-axis, TEM produces no change for a given wind speed of e.g., 10 m s$^{-1}$. The
cause of change in the TEM at 10 m s$^{-1}$ (solid red line) is due solely to the value of $E$ varying. Since $E$ is not aerodynamic,
dust emission does not change except when $E$ changes. Under a scenario with the wind speed reduced from 10 m s$^{-1}$ to 8 m s$^{-1}$, the TEM $F$ increases but at a reduced rate; that rate does not change with $u_{s*}/U_{10}$. Similarly, when the wind speed
increases from 10 m s$^{-1}$ to 12 m s$^{-1}$, the TEM $F$ increases at an increased rate, but does not change with $u_{s*}/U_{10}$.
In contrast, for a given wind speed of 10 m s$^{-1}$, the AEM produced the greatest reduction in dust emission with the greatest
decrease in $u_{s*}/U_{10}$ (the largest increase in roughness; **Figure 2b**). With the greatest increase in $u_{s*}/U_{10}$ (the largest
decrease in roughness) the largest increase in dust emission is produced by the AEM. When wind speed is consistently
reduced to 8 m s$^{-1}$, the change in dust is smaller with 10 m s$^{-1}$. Notably, there is no change in dust emission between a change
of -0.01$<u_{s*}/U_{10}>$0.01 (**Figure 2b**). When wind speed is consistently increased to 12 m s$^{-1}$, the change in dust emission
produced by the AEM is large, continuous and evident as $u_{s*}/U_{10}$ changes.
The results of these simulations illustrate how the TEM does not adequately represent vegetation sheltering
dynamics and that $E$ merely adjusts the magnitude, not the onset of dust emission. In contrast, the AEM provides a direct
estimate of $u_{s*}$, which modifies dust emission as roughness and / or wind speed changes. Since this direct estimate of $u_{s*}$ is
available from albedo, either monitored from satellite remote sensing or modelled prognostically in ESMs, it is available
over space and / or time without the need for $R$ or the bare soil fraction $E$, thereby reducing uncertainty in the model
parameterisation.

### 3.2 Modelled and observed dust emission frequency at DPS locations.

We reproduced DOD > 0.2 probability at previously identified DPS locations across southwestern areas North America to
compare with their observed frequency (**Figure 3**). The probability of DOD showed little resemblance to DPS, with a
distinctly different spatial pattern and considerably greater probability in some areas. Peak DOD occurred across the USA /
Mexico border in the Chihuahuan Desert, while DPS peaked over the Southern High Plains in eastern New Mexico and
western Texas. DOD probability increases in areas of reduced vegetation roughness (**Figure 1**) as difficulties in measuring





atmospheric dust over dark surfaces (e.g., vegetation), limit the DOD data to only the most arid regions. In areas where the data are comparable (e.g., northern Chihuahuan Desert; 108°-104°W, 29°-32°N), DOD probability is (at least) an order of magnitude greater than DPS.


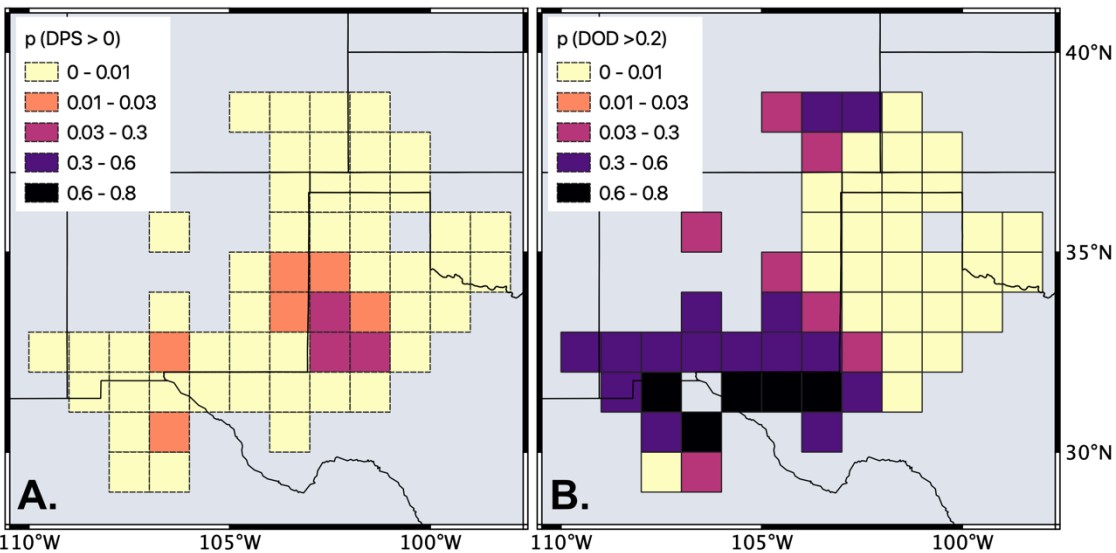

**Figure 3**. Comparison between the probability of observed dust emission point sources (DPS > 0) observations (a) and MODIS (b) dust optical depth (DOD > 0.2) during the period of DPS observation (2001-2016). All available MODIS DOD data were used, quality flags were not used to filter these data. The missing value of the pixel in the south-east of MODIS
DOD is evident in the original data and has not been removed during processing.

We compared estimated dust emission frequency (AEM and TEM models with *F* > 0 or DOD > 0.2) with observed DPS frequency (in days per year) at each DPS grid location (**Figure 1**). For each model comparison, the observed DPS frequency remained the same, with differences in the model described on the x axis (**Figure 4**). At most grid points,
modelled frequency exceeds observation. Both AEM and TEM over-estimate dust emission frequency with RMSE = 20 and 27 days per year respectively (**Figure 4**). Nevertheless, across all grid box data, the relation between DOD and DPS was very large exceeding DPS frequency by nearly 2 orders of magnitude, with RMSE = 151 days per year, considerably larger than the relation between DPS and the dust models. Least squares log-linear regression models were fitted to all models, with AEM and TEM frequencies showing significant correlation with DPS observed frequency, producing a regression slope
of 0.5 (AEM) 0.51 (TEM) and R$^2$ = 0.43 and R$^2$ = 0.48 (*P*<<.001). DOD frequency did not show a significant correlation with DPS observed frequency, with a regression slope of 0.07 and R$^2$ = 0.01, *P*.35, as shown in **Figure 3**.





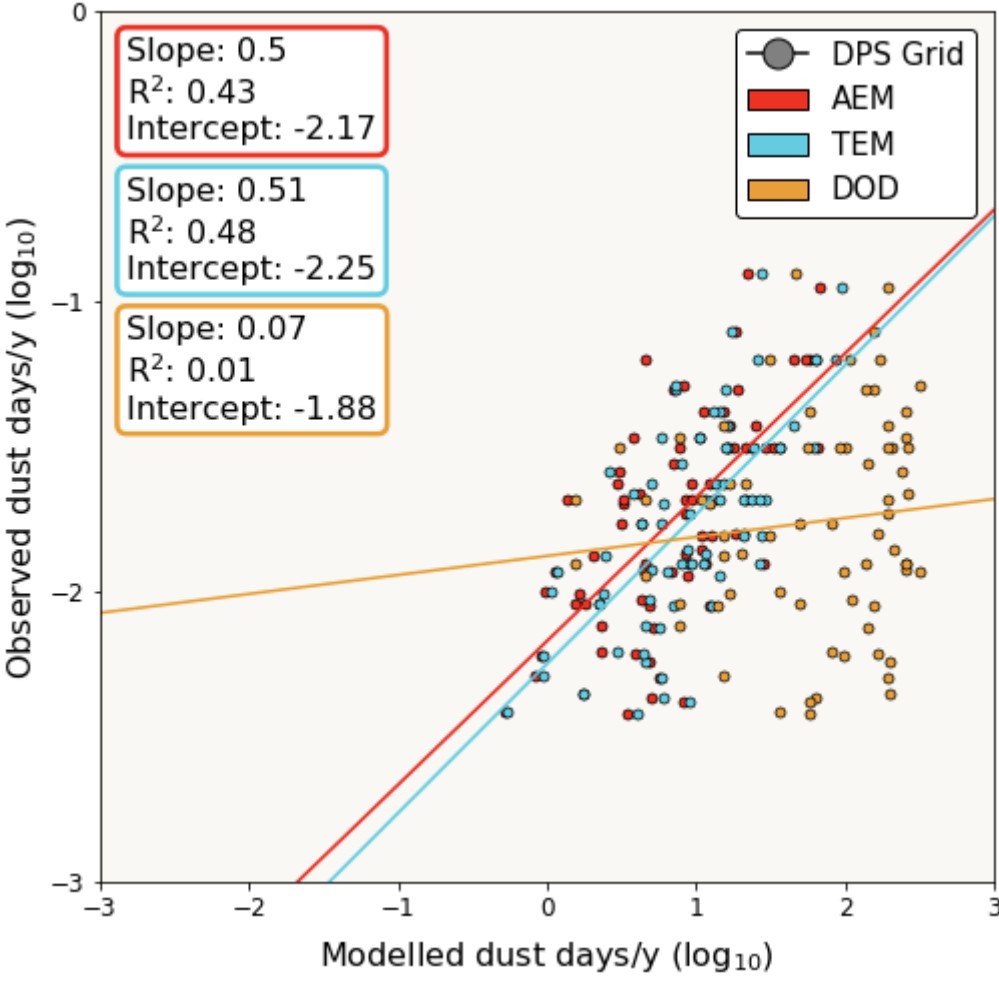

**Figure 4** Modelled and observed frequency at known North American satellite observed dust emission point sources (DPS), identified in satellite observations (Kandakji et al., 2020; Lee et al., 2012; Baddock et al., 2011). For each point, the y axis represents the observed number of DPS observations (per grid cell) per year during different observation phases of the DPS datasets within the observation time period (2001 – 2016). For AEM and TEM, the x axis describes number of modelled observations (F>0) at DPS locations in each grid cell per year during the same time period (x axis). For DOD, the x axis describes the frequency that DOD > 0.2 per year for the same period. The least squares logarithm regression of modelled against DPS observations produced the model parameter coefficients, R2 correlation and the square root of the mean squared difference between DPS, and model predictions (RMSE) adjusted by the degrees of freedom using the number of dust emission model parameters (df = 9 for AEM; 12 for TEM; 6 for DOD).





### 3.3 Modelling dust emission change over space and time

The mean $u_*/U_{10}$ and full range of $U_{10}$ for the year 2020 are shown (**Figure 5a & b**). For consistency with Figure 2, the mean dust emission is shown for selected wind speeds ($U_{10}$ = 8.5 – 9.5 m s$^{-1}$) from both AEM and TEM (**Figure 5c & 5d**). The

spatial distribution of mean dust emission varied between AEM and TEM in both magnitude and spatial extent of dust emission. According to AEM, large dust emissions (0.05 – 0.12 kg m$^{-2}$ y$^{-1}$) occurred in discrete areas across the Southern High Plains (104.5°W, 33.5°N), northern Chihuahuan Desert (107.5°W, 32°N), southwest Colorado Plateau (110.5°W, 35°N), and the Great Divide Basin within the Wyoming Basin (108.5°W, 42°N). These areas correspond with small $u_*/U_{10}$, and large wind speed ($U_{10}$). TEM dust emission occurred with similar magnitude over a greater area, including large parts of

New Mexico and Wyoming, while also extending through the Great Plains in northwest Texas, Oklahoma, Colorado, and Nebraska (**Figure 5d**). This pattern matches closely the distribution of mean $U_{10}$ (**Figure 5b**).





**Figure 5.** Mean conditions for North America during the year 2020 for peak dust season months March-May, including (a) total wind friction velocity ($u_*$) scaled by wind speed at 10m height ($U_{10}$), (b) wind speed, and modelled dust emission with (c; AEM) and without (d; TEM) varying aerodynamic roughness. The dust emission displayed is for wind speeds restricted to between 8.5-9.5 m s$^{-1}$ (for comparison with **Figure 2**). The daily maximum of hourly data from ERA5-Land (Source: ECMWF) are used in both models.

Differences in mean dust emission during peak dust season (MAM) for years 2001 and 2020 greater than MDC significance ($P < 0.05$) were produced for both TEM and AEM (**Figure 6c & d**). These were compared to total mean difference in $u_*/U_{10}$ and $U_{10}$ during the same periods (**Figure 6a & b**). Comparing the change ($\Delta$) between the two periods, $\Delta u_*/U_{10}$ across North America produced a range +/- 0.01, with the greatest reduction ($< -0.01$) associated with decreased roughness in Canada, very likely caused by changes in snow coverage. Note that snow is removed from $u_*/U_{10}$ when calculating dust emission.





South of the USA/Canada border, roughness reduced (-0.01) across large areas of Montana, the Wyoming Basin, and eastern

parts of the Great Plains (Colorado, Kansas, and Nebraska). Further reductions in $u_*/U_{10}$ (-0.01 to -0.005) occurred in discrete areas of the Southern High Plains, and northern Chihuahuan Desert. The greatest increase in $u_*/U_{10}$ (> 0.01) occurred across the American Mid-West, including Minnesota, Iowa, and South Dakota. In dusty areas (**Figure 5**), the greatest increase in $u_*/U_{10}$ (0.005 to 0.01) occurred as discrete locations within the Chihuahuan and Sonoran Desert, the Great Basin (Nevada), and the southern extent of the Southern High Plains (eastern New Mexico and western Texas). Mean

$\Delta U_{10}$ produced a range +/- 1.6 m s$^{-1}$, with the largest increase (>1.6 m s$^{-1}$) across southwest USA, including the Great Basin, Mojave and Sonoran Deserts and the Colorado Plateau. Mean $U_{10}$ reduced (<-0.8 m s$^{-1}$) in the Mid-West states of Wisconsin and Illinois.

Between 2001 and 2020, significant change in dust emission (D$F$) from AEM and TEM varied across the range +/- 2 kg m$^{-2}$ y$^{-1}$. AEM produced a significant decrease in $F$ (-1 to -2 kg m$^{-2}$ y$^{-1}$) from several areas, including the Southern High

Plains (eastern New Mexico and western Texas), the Colorado Plateau, and the Sonoran Desert (**Figure 6c**). The AEM showed a significant increase in $F$ from the Wyoming Basin, and discrete locations in Montana, and western areas of the Great Plains (west Colorado, Nebraska). In contrast, where no change in the AEM was detected, the TEM produced a significant decrease of $F$ during the 2020 period across large areas of the Great Plains (up to -2 kg m$^{-2}$ y$^{-1}$), the arid southwest (-1 to -2 kg m$^{-2}$ y$^{-1}$), including the Mojave, Sonoran, and Chihuahuan Deserts, and the Mid-West (-1 to -2 kg m$^{-2}$ y$^{-1}$

$^{-1}$). The TEM $F$ increased significantly across the Wyoming Basin (up to 2 kg m$^{-2}$ y$^{-1}$), the Great Basin and northern Mexico (**Figure 6d**).

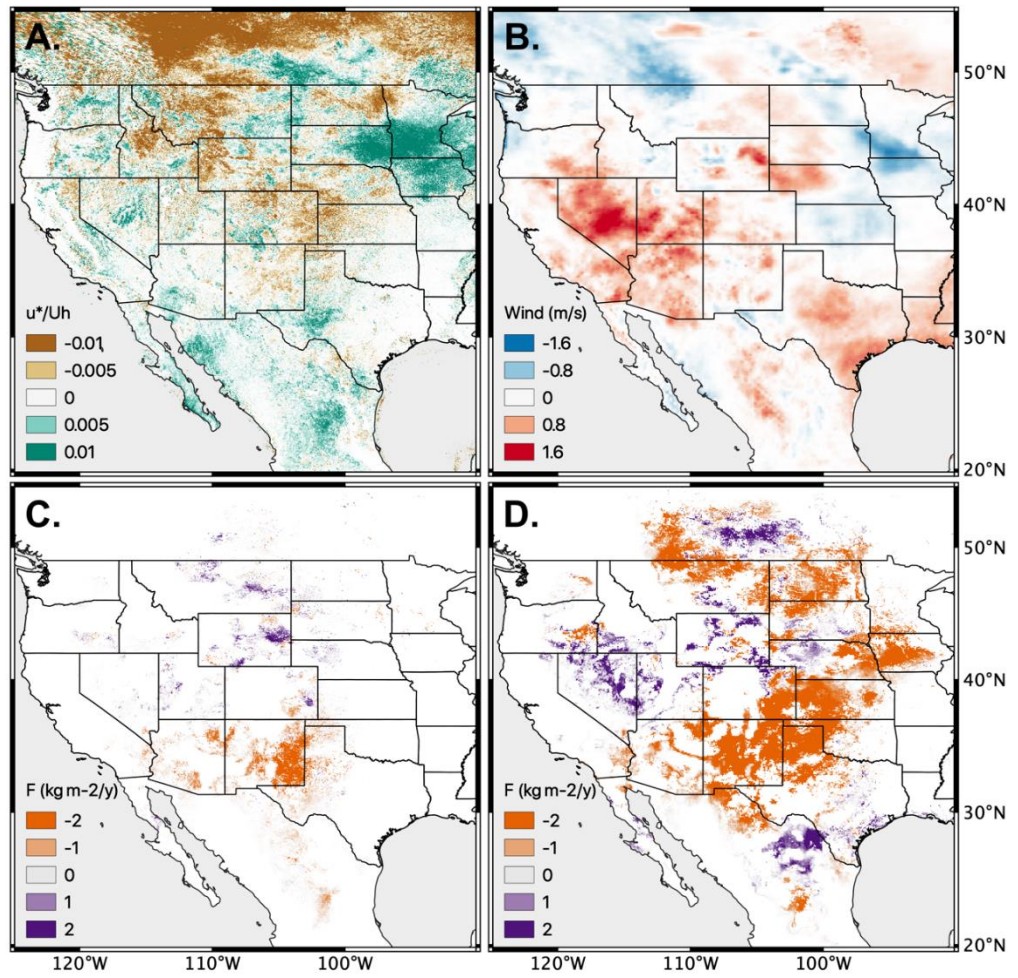

**Figure 6**. Difference maps between the year 2001 and the year 2020 for the peak dust season months March-May and only dust days (not all days), showing total difference in (a) mean wind friction velocity ($u_*$) scaled by wind speed at 10m height ($U_{10}$) and (b) wind speed ($U_{10}$). Minimal detectable change in dust emission with significance ($P > 0.05$) with AEM varying aerodynamic roughness (c) and with TEM $z_0$ fixed and static over time (d). Wind data = ERA5-Land (Source: ECMWF). See Appendix for details on the calculation of the minimum detectable change.

## 4. Discussion

### 4.1 Overcoming dust emission model weaknesses using the albedo-based approach

Dust emission modelling has historically struggled to represent adequately soil wind friction velocity. Many of the TEMs assume homogenous bare ground, before using the complement of vegetation cover to reduce emission. Using satellite observed dust emission point sources (DPS; **Figure 1**) we have shown that TEMs overestimate dust emission frequency by





nearly an order of magnitude (RMSE = 0.76 using $\log_{10}$) (**Figure 4**). Using albedo to describe variability in aerodynamic roughness through changes in vegetation structure, the AEM performs theoretically better (**Figure 2**) at correctly estimating

the probability of $u_{s*}$ exceeding the entrainment threshold, and subsequent changes in dust emission timing and magnitude. When compared to observed DPS (**Figure 4**), AEM performs only moderately better than TEM, still overestimating dust emission frequency by 0.6 orders of magnitude (RMSE = 0.6 using $\log_{10}$). However, it is important to recall that the AEM is not tuned in any way, but the TEM is tuned using values of z0m an z0s which are fixed over space and static over time and then dust emission is adjusted by $E$. Furthermore, most DPS are from predominantly barren and windy environments, with

mean $u_*/U_{10}$ of 0.069 and mean $U_{10}$ of 6.9 m s$^{-1}$, reducing the potential influence of dynamic vegetation. Nevertheless, the over-estimation of dust emission caused by the frequency of occurrence being too large relative to the observed frequency occurs because of one of more of the factors described in **Table 1**. Those factors are classified to form a future research priority based on the results and conclusions reached in this study and based on our understanding of the process that has arisen during the investigation of the results.


**Table 1**. Assessment of the factors causing over-estimation of dust emission frequency, their likely impact on dust emission modelling and suggested priority for research investment.

| Factors causing over-estimated dust emission frequency | Assessment of impact on dust emission modelling | Research priority |
|---|---|---|
| Modelled $u_{*ts}$ at the grain scale is very likely to be much smaller in value than that of $u_{*ts}$ at 500 m (MODIS albedo). The generalized problem is that $u_{*ts}$ is not upscaled for use with $u_{s*}$ and the (typically larger) scale of wind speed data (see below). The modelled $u_{*ts}$ is also assumed fixed over space and static over time. | The scale difference is very likely causing $u_{*ts}$ to be too small relative to $u_{s*}$ causing $u_{*ts}$ to be exceeded too frequently and hence over-estimating dust emission (too many dust days). | High |
| Dust emission modelling assumes an infinite supply of dry, loose erodible material is available once $u_{*ts}$ has been exceeded. | Under this assumption, the amount of dust emission which occurs when $u_{*ts}$ is exceeded is over-estimated where sediment is unavailable and / or restricted by rocks and biogeochemical soil crusts. | High |
| Modelled $U_{10}$ may be too large. However, the scale-invariant albedo-based approach (Ziegler et al., 2020) able to operate over large grid boxes should eradicate scale differences. | Wind speed may be too large despite the considerable effort to reproduce realistic wind fields. Tied to the evaluation of wind speed magnitude for dust emission, is the discrepancy between the scale of wind fields e.g., ERA5-Land | Medium |





| | 11 km pixels, and the scale of dust emission modelling e.g., MODIS 500 m pixels. | |
|---|---|---|
| The DPS are derived from polar-orbiting satellite observations, which may not accurately and completely identify the sources and frequency of all dust emissions. | The grid-boxes approach used here overcome concerns about precision in the location of dust points. Inevitably, there is a scale dependency to the frequency of occurrence that needs to be quantified. Small magnitude, high frequency dust emissions may not be included in the observed dust emissions at point source (DPS). | Medium |
| The wind tunnel data used in the albedo-based drag partition calibration may not represent the complete range of conditions and flexibility in the vegetation (deforming to change shape). | This research hypothesis queries the calibration but, given the nature of the data already included probably amounts to reducing the uncertainty in the calibration. | Low |

Here, we use the latest version of ERA5-Land wind (at 10 m height) data at a reasonably fine (11 km) resolution. It is evident that $U_{10}$ is over-estimated in some regions (Fan et al., 2021). However, there appears to be no systematic bias that would lead to the over-estimation of dust emission frequency. The grain scale of $u_{*ts}$ is evidently incompatible with dust emission modelling over area (e.g., pixels at 500 m), and this factor appears to be the most likely cause of the over-estimated model dust emission frequency and should be a priority for future work. Without resolving the scale of $u_{*ts}$ it is not possible to determine the impact of the assumed infinite supply of loose erodible material (**Table 1**). It is very likely that these two factors explain the first-order differences between the DPS frequency and the dust emission model frequency. There remains uncertainty over the use of DPS frequency. However, by comparison with dust in the atmosphere represented by DOD, the use of DPS frequency is up to two orders of magnitude smaller. There is a small, perhaps lower-order likelihood that the original calibration of the albedo-based approach is not representative and universal, despite recent support for the approach (Ziegler et al., 2020).

Beyond these observed dust emission point sources, vegetation roughness appears more influential, constraining dust emission greater than 0.1 kg m$^{-2}$ a$^{-1}$ to areas where $u_*/U_{10}$ is no greater than 0.06, even during periods of peak (8.5 – 9.5 m s$^{-1}$) wind speed. In contrast, TEM predicts dust emission >0.1 kg m$^{-2}$ a$^{-1}$ in areas where $u_*/U_{10}$ is greater than 0.075, including large areas of the Great Plains. This difference is emphasized in parts of western Oklahoma (99.5°W, 35.5°N), where mean $u_*/U_{10} > 0.08$ prevent dust emission from the AEM, despite a mean $U_{10} > 7$ m s$^{-1}$. However, in those areas TEM dust emission exceeds 0.2 kg m$^{-2}$ a$^{-1}$. These contrasting estimates emphasise TEM dependency on variability in $U_{10}$, due to the use of $u_*^3$ and the inability of $R(z_0)=0.91$ fixed over space and time to correctly attenuate wind speeds by aerodynamic roughness. This limitation creates two main issues, a) a requirement for post-process tuning, which restricts model ability (or





increases uncertainty) to effectively predict dust without *a priori* information; b) large scale uncertainty driven by a large spatial and temporal variability in $U_{10}$.

### 4.2 Overcoming dust emission model tuning to dust in the atmosphere

Previously, inconsistency in modelled dust emission from areas unlikely to produce dust has been filtered out by utilizing a preferential dust source map (Ginoux et al., 2012), whereby the probability of dust emission is pre-defined by the topographic setting, constraining emission to drainage basins (Zender et al., 2003b). These pre-defined conditions limit the ability to simulate the dynamics of dust emission in these areas, as well as omitting most small dust sources in other areas of the basin (Urban et al., 2018). Furthermore, modelled dust emission frequency is typically several orders of magnitude greater than observation, creating the need for calibration when integrated into ESM. Currently, a global observed dust emission archive does not exist, thus calibration is achieved against observed dust in the atmosphere (e.g., DOD). However, we have shown that DOD poorly represents observed dust emission frequency by nearly two orders of magnitude, with no spatial correlation in frequency variability. Previous studies have suggested that this inconsistency is due to the spatial bias between time of emission and downwind observation in sun-synchronous daily observations (Schepanski et al., 2012). Whilst explaining some of the inconsistency in our results, it also illustrates the fundamental problem of calibrating dust emission using dust in the atmosphere. Using extant DPS, our results demonstrate that DOD is limited to areas with highly reflective surface e.g., creating a bias over northern areas of the Chihuahuan Desert. The DOD hotspots for the period 2001-2016 were located upwind of the DPS locations. These findings severely undermine the efficacy of dust emission model calibration to DOD, especially in areas where dust emission occurs in relatively discrete areas surrounded by more densely vegetated areas such as North America. Over-estimation of dust emission in these environments very likely alters the magnitude and frequency of the global dust distribution, which currently considers continental-scale barren environments (e.g., North Africa, Middle East) as the main source of dust.

Our comparison of dust emission between two time periods emphasizes a previously unrealised impact of dynamic aerodynamic roughness in the temporal variability of dust emission magnitude. Through the correct calculation of $u_{s*}$, the AEM constrains dust emission to relatively small areas, restricting significant variability between time steps to only dust producing areas (e.g., the arid southwest and semi-arid parts of the Great Plains - **Figure 6c**). In contrast, TEM's dependency on $U_{10}$ variability produces significant changes in dust emission over vast vegetated areas, including those which are unlikely to produce dust (e.g., temperate areas of the Great Plains and the grasslands of North Mexico; **Figure 6d**).

### 4.3. Implications of model deficiencies for dust emission modelling

Our study has demonstrated that dust emission modelling can be considerably improved by utilising a calibrated drag partition, rather than the traditional static approach. The TEMs were developed more than two decades ago when dynamic data inputs were less available. Many global dust emission studies still use static inputs, such as vegetation cover thresholds and bare soil fraction in global dust emission modelling (Albani et al., 2014). Preferences for which regions emit or how





much vegetation to allow before dust emission ceases, have contributed to the inability to detect model weaknesses (Zender et al., 2003a). The *ad hoc* delineation of source regions and / or tuning to dust in the atmosphere, constrains dust emission to areas with large concentrations of dust in the atmosphere (Huneeus et al., 2011). However, there may be regional differences in magnitude and frequency of dust emission, wind speed and particle size controlling dust residence times. Furthermore, current atmospheric dust loads do not enable unbiased reconstruction of past trends or to project future shifts in the location

or strength of emissions (Mahowald et al., 2010). There is also a great risk that the major scientific advances made in developing dust emission schemes (Marticorena and Bergametti, 1995; Shao et al., 1996) and newly developed data / parameterizations (Prigent et al., 2012) are being overlooked by an over-reliance on simplistic assumptions about dust source location and erodibility to implement dust emission models. Model 'tuning' in its various guises, makes it hard to routinely evaluate the dust emission implementation. We contend that it is essential to ensure that the balance of dust emission

modelling is towards the fidelity of the dust emission scheme (processes) rather than the parsimony of its implementation (parameterization) (Raupach and Lu, 2004). As new parameterization schemes are developed and new data sources become available, the research community will benefit from being open to critical re-evaluations to avoid model deficiencies enduring.

Incomplete TEMs predict unreasonably large dust emission particularly in vegetated regions, because $u_{s*}$ is over-

estimated. Despite their multiple parameters, incomplete TEMs operate like other dust emission models explicitly controlled only by wind speed (e.g., GOCART) at some $f$ height $U_f$ and $t$ threshold of $U_{ft}$ (Ginoux et al., 2001). In our study, we did not include these dust emission models based on wind threshold. However, given their similarity with the incomplete TEMs, our results suggest that both of these model types are inadequate for representing dust emission across Earth's dynamic vegetated drylands and over time. Model weaknesses most likely explain why on monthly time scales, the relation between

surface wind speed and TEMs could be linearized, and why differences between CMIP5 models appear to be due solely to wind field biases (Evan et al., 2016). Perhaps most significantly, our results explain to a large degree how the incomplete TEMs lack validity in 21st century dust emission projections (Evan et al., 2014).

**5 Conclusion**

Improving climate change projections requires dust models that are sensitive to and accurately represent dust emission

responses to changing environmental conditions (wind speed, precipitation, evapotranspiration), land use and land cover dynamics. The incomplete TEMs were shown here to over-estimate dust magnitude, frequency and extent, and lacks the dynamics in dust emission of the albedo-based approach. Albedo is increasingly available from accurate and precise ground measurements using net radiometers, from various airborne and satellite platforms most notably MODIS and more recently VIIRS, or intrinsic prognostic estimates used in ESMs. The use of albedo as a prognostic variable provides the opportunity

for this new albedo-based approach to be readily adopted in ESMs. Therefore, coupling the albedo-based approach to ESMs is expected to reduce uncertainty in dust emission and may transform climate change projections.





## 6. Code Availability

The Google Earth Engine Java script code is available to run using the links below for the traditional dust emission model (TEM) and the albedo-based dust emission model (AEM).

TEM - https://code.earthengine.google.com/97aaaad02da2af9b914fff8d9cd1bf5d

AEM - https://code.earthengine.google.com/9726348d2fc3e81381e8a9229667afdd

The code is archived as a text file using Zenodo (where the code will not run without access to the Google Earth Engine) using the DOI below

https://doi.org/ 10.5281/zenodo.5626825

## 465    7. Data Availability

The data used are identified in the main text and below using the the Google Earth Engine data description and catalogue references, link and DOI.

| Dates used | Google Earth Engine data | Google Earth Engine Catalogue reference, link or DOI |
|---|---|---|
| 2009 | MODIS land cover used to mask land / sea | MODIS/051/MCD12Q1/2009_01_01 https://doi.org/10.5067/MODIS/MCD12Q1.006 |
| Static | ISRIC clay content | https://github.com/ISRICWorldSoil/SoilGrids250m/ |
| 2001-2020 | MODIS albedo (Band1 Band1_iso) | MODIS/006/MCD43A1 https://doi.org/10.5067/MODIS/MCD43A1.006 |
| 2001-2020 | ECMWF ERA5-Land u-component_of_wind_10m v-component_of_wind_10m volumetric_soil_water_layer_1 soil_temperature_level_1 | ECMWF/ERA5_LAND/HOURLY doi:10.24381/cds.e2161bac |
| 2001-2020 | MODIS Snow Cover | MODIS/006/MOD10A1 https://doi.org/10.5067/MODIS/MOD10A1.006 |
| 2001-2020 | MODIS Normalised Difference Vegetation Index | MODIS/MOD09GA_006_NDVI https://doi.org/10.5067/MODIS/MOD09GA.006 |





## 8. Appendix

### 8.1 Implementation of traditional dust emission scheme (TEM)


When the dust emission scheme (Marticorena and Bergametti, 1995) is implemented (Eq 2 and 4), the wind friction velocity ($u_*$) is assumed to be the total $u_*$. We set $c$=1, the air density was fixed for simplicity ($\rho_a$=1230 g m⁻³). The acceleration due to gravity was also fixed ($g$=9.81 m s⁻²). Following the current practice, we calculated $u_*$ from the modeled 10 m wind velocity using the logarithmic layer profile theory and aeolian roughness length (Darmenova et al., 2009) following the Monin-Obukhov similarity theory:


$$u_* = \frac{kU_f}{\ln\left(\frac{Z_U}{Z_0}\right)+\varphi_m},$$ (Eq. 6)

where $\varphi_m$ is the stability function accounting for a deviation of the wind profile from the logarithmic, von Kármán's constant ($k$=0.4) and $Z_U$=10 m the height at which the freestream wind speed $U_{10}$ estimates were provided. We assumed the wind profile is logarithmic and stable and used modelled wind speed (10 m) data from the ECMWF ERA5-Land (Muñoz Sabater, 2019) (hourly; ~11 km).


Estimates of the aerodynamic roughness length ($z_0$) were fixed over time and fixed over space. The threshold of sediment flux ($u_{*t}$) is commonly represented as only an energy-limited process by calculating it as:


$$u_{*t}(d, w, Z_0, Z_{0s}) = \frac{u_{*ts}(d)H(w)}{R(Z_0, Z_{0s})},$$ (Eq. 7)

where the entrainment threshold $u_{*ts}(d)$ for a given size fraction $d$, can be modified by functions including the drag partition $R(z_0, z_{0s})$ and the moisture content $H(w)$. The $u_{*ts}$ of a given $d$ (mm):


$$u_{*ts}(d) = \begin{cases} \frac{0.129K}{(1.928Re^{0.092}-1)^{0.5}}, & 0.03<Re\leq10 \text{ or } Re>10, \\ 0.129K(1-0.0858)e^{-0.0617(Re-10)} \end{cases}$$ (Eq. 8)

$$Re = aD^x + b; a = 1331cm^{-x}; b = 0.38; x = 1.56,$$ (Eq. 9)

$$K = \left(\frac{\rho_p gd}{\rho_a}\right)^{0.5}\left(1+\frac{0.006}{\rho_p gd^{2.5}}\right)^{0.5},$$ (Eq. 10)


includes $p_a$=1230 g m³ fixed air density, $p_p$=2650 g m³ fixed particle density, $g$=9.81 m s⁻² acceleration due to gravity. The dimensionless function $H$ (Fécan et al., 1998) was developed using wind tunnel experiments to account for gravimetric





surface soil moisture content $w$ (kg$^3$ kg$^{-3}$) using the difference between the potential $w'$ based on clay content and near surface $w$:

$$H(w) = \sqrt{1 + (1.21(w - w')^{0.68})} \qquad \text{(Eq. 11)}$$

where

$$w' = 0.0014\%clay^2 + 0.17\%clay , \qquad \text{(Eq. 12)}$$

and clay is the finest fraction (expressed as a percentage) of the soil and typically less than 2 µm.

A discussion of the use of this parameterization in dust emission schemes is included elsewhere (Zender et al., 2003a; Xi and Sokolik, 2015). We make use of the ERA5-Land volumetric soil moisture data (0-7 cm of soil layer; hourly; 11 km). To convert from volumetric soil moisture to the required gravimetric soil moisture we divided by the soil bulk density. We assumed that the gravimetric moisture of the uppermost soil layer was 20% of the 7 cm soil layer (Edwards et al., 2013). The soil bulk density and clay, silt and sand soil texture are from ISRIC (Hengl et al., 2017) and is fixed over time (250 m).

The $R(z_0, z_{0s})$ is valid for small wakes ($z_0 < 1$ cm), and to parameterize solid obstacles only. This poses a problem in applying this approach to partially vegetated surfaces such as mixed grasslands, shrublands, and agricultural/shrubland mosaics (Darmenova et al., 2009). Applying different parameterizations for surfaces with similar roughness values could result in a significant discrepancy in the estimated drag partition (Darmenova et al., 2009). To reduce the impact of this discontinuity on $R(z_0, z_{0s})$, a modification is used because it includes a wider range of land surface types


$$R(Z_0, Z_{0s}) = 1 - \frac{ln\left(Z_0/Z_{0s}\right)}{ln\left[0.7\left(12255cm/Z_{0s}\right)^{0.8}\right]}. \qquad \text{(Eq. 13)}$$

In the absence of regional and global spatio-temporal dynamics of $R$ and aerodynamic roughness length ($z_0$) data to calculate $u_*$ from $U_{10}$, two approaches for representing surface roughness have been developed in regional and global dust emission modelling over the last two decades. The older, but still common approach uses globally constant values of $z_0$, fixed over

time (Zender et al., 2003a; Ginoux et al., 2001; Mahowald et al., 2010; Woodward, 2001; Tegen et al., 2002). Fixed aerodynamic roughness length for the landscape $z_0$=100 µm and the soil $z_{0s}$ = 33.3 µm, fixes $R(z_0) \approx 0.91$ which assumes that the Earth's land surface is devoid of vegetation roughness and static over time. This approach therefore tends to over-estimate dust emission. With $R(z_0)$ fixed, $R(z0)u_* = u_{s*}$ is assumed. We recognize that the use of a constant value for $z_{0s}$ smooths out the heterogeneity of dust sources. We also know that it is recommended to use a $z_{0s} \approx 1/30$ of the coarse mode

mass median diameter of the undisturbed soil size distribution, instead of setting it to a fixed constant (which assumes that the coarse population of an undisturbed soil is equivalent to the coarse population of the soil texture (Darmenova et al., 2009). Nevertheless, we fixed $z_{0s}$ to ensure that results were consistent with previous work. A second approach is to use spatially heterogeneous estimates for arid and semi-arid regions (Prigent et al., 2012). That work follows continued efforts to



use active and passive reflectance obtained from satellite remote sensing to characterize aerodynamic roughness(Marticorena

et al., 2006). Although this approach provides an observation-based approximation of $z_0$, it remains a challenge to estimate $R$

to approximate $u_{s*}$ necessary for the complete sediment flux equation.

To implement vertical dust emission, we introduced additional terms to Eq. 5 which are explained below

$$F = A_n A_f E M_i(d) Q_i(d) 10^{(0.134 clay_\% - 6.0)}. \qquad \text{(Eq. 14)}$$


Notably, no global tuning is applied to either the traditional or new albedo-based dust emission model. In the traditional dust

emission, we fixed the mass fraction of clay particles in the parent soil to $clay_\% = 20$ consistent with previous work (Zender et

al., 2003a). The proportion of emitted dust in the atmosphere $M$ for a given particle size fraction ($d$) is dependent on the

particle size distribution. We calculated the relative particle size surface area (Marticorena and Bergametti, 1995) ($M$). The

parameter $E$ was defined in the main text assuming $E=1-A_v$ so that vegetation indices can be used (Shao et al., 1996). To

conform with that practice, we calculated

$$A_v = -22.5 + 150 NDVI \qquad \text{(Eq. 15)}$$

from global daily $NDVI$ from MODIS (MOD09GA Collection 6 from Terra at 500 m pixel).

When the soil is covered by snow it is unable to provide any dust emission. In this situation it is most effective to use a mask

which determines whether snow is present or absent ($A_n$). However, the coverage of snow in a given pixel is an areal

quantity and therefore ranges between 0-1. Consequently, we applied the MODIS Normalised Difference Snow Index (Hall,

2016) (MOD10A1 from Terra, daily at 500 m). Similarly, if the soil is bare but frozen it is unable to release sediment almost

regardless of how much wind energy is applied. In this situation it is most effective to use a mask which determines whether

the soil is frozen or not ($A_f$). We used soil temperature available in ERA5-Land and set a threshold of 273.15 K above which

sediment flux can occur.

**8.2 New parameterization of $u_{s*}$ by relating shelter to shadow (AEM)**

To implement Eq. 1, we assume that the total wind friction velocity ($u_*$) is used in the sediment flux equation. We use a new

albedo-based implementation of the sediment flux equation which avoids $u_{s*}=u_* R$ and therefore does not use $u_*$, $R$ or the

aerodynamic roughness length of vegetation ($z_0$) or that of the soil ($z_{0s}$). Instead we used a robust direct estimation (Chappell

and Webb, 2016) for $u_{s*}$

$$\frac{u_{s*}}{U_f} = 0.0311 \left( exp \frac{-\omega_{ns}^{1.131}}{0.016} \right) + 0.007, \qquad \text{(Eq. 16)}$$



where $\omega_{ns}$ is the normalised and rescaled albedo ($\omega$) translated and scaled ($\omega_n$) from a MODIS range ($\omega_{n\,min}=0$, $\omega_{n\,max}=35$) for a given illumination zenith angle ($\Theta=0°$) to that of the calibration data ($a=0.0001$ to $b=0.1$) using the following rescaling equation (Chappell and Webb, 2016):


$$\omega_{ns} = \frac{(a-b)(\omega_n(\theta)-\omega_n(\theta)_{max})}{(\omega_n(\theta)_{min}-\omega_n(\theta)_{max})} + b. \tag{Eq. 17}$$

Shadow is the complement of albedo $1 - \omega_{dir}(0°, \ \lambda)$ and the spectral influences due to e.g., soil moisture, mineralogy and soil organic carbon, were removed by normalizing (Chappell et al., 2018) with the directional reflectance viewed and

illuminated at nadir $\rho(0°, \ \lambda)$:

$$\omega_n = \frac{1-\omega_{dir}(0°, \ \lambda)}{\rho(0°, \ \lambda)} = \frac{1-\omega_{dir}(0°)}{\rho(0°)}. \tag{Eq. 18}$$

This was implemented by making use of the available MODIS black sky albedo (Schaaf, 2015) to estimate $\omega_n$, and the

shadow is normalized by dividing it by the MODIS isotropic parameter $f_{iso}$ (MCD43A1 Collection 6, daily at 500 m) to remove the spectral influences:

$$\omega_n(0°) = \frac{1-\omega_{dir}(0°,\lambda)}{f_{iso}(\lambda)} = \frac{1-\omega_{dir}(0°)}{f_{iso}}. \tag{Eq. 19}$$

The $f_{iso}$ is a MODIS parameter that contains information on spectral composition as distinct from structural information (Chappell et al., 2018). In theory, the structural information is waveband independent(Chappell et al., 2018). The normalization of MODIS data using this parameter and that of MODIS Nadir BRDF-Adjusted Reflectance (NBAR) is similarly sufficient to remove the spectral content for all bands examined (Chappell et al., 2018). In practice, we calculated $\omega_n$ using MODIS band 1 (620-670 nm).

To calculate the vertical dust emission, we followed the same approach as above (Eq. 14) except for $E$ which was not used. In the new albedo-based model we used the latest, reliable spatially varying layer of particle size (Dai et al., 2019) and restricted $clay_\%$ to a maximum value of 20% consistent with reasonable results when applied in regional models (Woodward, 2001). This new implementation provides a highly dynamic representation of the soil wind friction velocity. To this model, we applied no other tuning.





### 8.3 Minimum detectable change framework


This approach, well-established for environmental resource monitoring (Chappell et al., 2015; De Gruijter, 2006; Webb et al., 2019), aims to establish the mean difference ($\hat{\bar{d}}_{2,1}$) of estimated means $\hat{\bar{z}}(t_1)$ and $\hat{\bar{z}}(t_2)$ between events $t_1$ and $t_2$ by

$$\hat{\bar{d}}_{2,1} = \hat{\bar{z}}(t_2) - \hat{\bar{z}}(t_1). \tag{Eq. 20}$$

The locations are pixels which are assumed fixed in space and are revisited over time. This static synchronous pattern
implies that in estimating the sampling variance of the change, a possible temporal correlation between the estimated means $\hat{\bar{z}}(t_1)$ and $\hat{\bar{z}}(t_2)$ must be taken into account. The true sampling variance equals

$$V(\hat{\bar{d}}_{2,1}) = V(\hat{\bar{z}}(t_2)) + V(\hat{\bar{z}}(t_1)) - 2\rho(\hat{\bar{z}}(t_2), \hat{\bar{z}}(t_2)), \tag{Eq. 21}$$

where $\rho$ is the temporal correlation between the two estimated means. As $\rho$ increases, the sampling variance of change gets smaller.

600        Our target quantity $\hat{\bar{d}}_{2,1}$ is greater than zero and statistically significant and defined as (Woodward, 1992):

$$H_0: \hat{\bar{z}}(t_1) = \hat{\bar{z}}(t_2),$$

$$H_1: \hat{\bar{z}}(t_1) = \hat{\bar{z}}(t_2) + \theta \; (\theta \neq 0). \tag{Eq. 22}$$

The alternative hypothesis $H_1$ is the adjustment due to $\theta = \hat{\bar{d}}_{2,1}$ which between sampling periods $t_1$ and $t_2$ is the net result of change in the property of interest during an intervening time. The uncertainty due to reaching an incorrect conclusion is the
minimum detectable change (MDC) which is related to the probability of the errors on the conclusion. In general, the smaller the MDC, the larger the required sample size for a given probability of false acceptance error (De Gruijter, 2006).

Our $H_0: \hat{\bar{d}}_{2,1} = 0$ is that the average difference in our property of interest has stayed the same over time. The alternative hypothesis $H_1: \hat{\bar{d}}_{2,1} \neq 0$ is that the average difference in our property of interest has changed over time. In statistical hypothesis testing two types of errors may be made. We may reject $H_0$ and conclude that there is a positive effect
when in reality there is no effect (false rejection; type-I error). We assigned a probability denoted $\alpha$ to this type of error and decide on a value of 5% based on the implications of making a false rejection. The alternative error is that we may accept $H_0$ and conclude that there is no effect, when in reality there is a positive effect (false acceptance; type-II error, $\beta$). The probability $1 - \beta$ is referred to as the power of the test and is used as a quality measure with a value set at 5%. First the critical value is calculated for the mean beyond which $H_0$ is rejected. The power is the probability that one correctly
concludes that there is a positive effect, that $\hat{\bar{d}}_{2,1} \neq 0$. The power of the test depends on $\hat{\bar{d}}_{2,1}$ i.e., the greater $\hat{\bar{d}}_{2,1}$, the larger the power.

A two-tailed test (for change without direction) statistic is commonly based on the t-test (Woodward, 1992):

$$(X_{1-\alpha} + X_{1-\beta})^2 = \frac{\hat{\bar{d}}_{2,1}^2}{\frac{V(\hat{\bar{z}}(t_1))}{N_1} + \frac{V(\hat{\bar{z}}(t_2))}{N_2}}, \tag{Eq. 23}$$




where $X$ is a standard normal distribution. Re-arranging to give an expression for $\hat{\bar{d}}_{2,1}$, that is the difference between means

which it is possible to detect with the specified power (and size) of test or more usefully, the smallest difference detectable with at least the given power

$$\hat{\bar{d}}_{2,1} = \left(X_{1-\alpha} + X_{1-\beta}\right)\left(\frac{V\left(\hat{\bar{z}}(t_1)\right)}{N_1} + \frac{V\left(\hat{\bar{z}}(t_2)\right)}{N_2}\right)^{0.5}. \tag{Eq. 24}$$

This last equation is our estimate of the difference in means and our MDC for a given set of conditions which were applied to our properties of interest.

## 9. Author Contributions

AC coded the dust emission schemes, MH coded the data analysis and AC and MH performed the analysis jointly. NPW co-wrote the first draft of the manuscript with AC. All authors contributed to revisions of the manuscript and development of the figures to form the final submission.

## 10. Competing interests

The authors declare that they have no conflict of interest.

## 11. Acknowledgments

The first author is grateful to Google for access to and use of the Google Earth Engine (GEE) and coding support from Noel Gorelick and coding advice from GEE forum members. We thank the following people for their specialist advice on earlier drafts of the manuscript: Beatrice Marticorena and Giles Bergametti, LISA; Amato Evan, Scripps Institution of
Oceanography; Stephanie Woodward and Malcolm Brooks, UK Met Office; Paul Ginoux, NOAA; Jasper Kok, UCLA; Natalie Mahowald, Cornell University; Ian Hall and Huw Davies, Cardiff University. We thank the following organisations for the use of their data: National Centers for Environmental Prediction (NCEP), NOAA AVHRR Surface Reflectance product; NASA EOSDIS Land Processes Distributed Active Archive Center (LP DAAC), USGS/Earth Resources Observation and Science (EROS) Center, Sioux Falls, South Dakota; ISRIC SoilGrids; The work was produced whilst AC
and NPW were funded by a joint grant from the National Science Foundation and Natural Environmental Research Council (EAR-1853853).

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
