# Peer review of "Weaknesses in dust emission modelling hidden by tuning to dust in the atmosphere"

_Geoscientific Model Development, 2021_

## Community Comment (CC1)

Dear Adrian and co-authors,
I made below a few comments related to your submitted manuscript as I found several cases of mis-representation or inaccurate description of the model and data I have been developing with my collaborators. Hopefully, you will find them useful to improve the manuscript.
Regards,
Paul Ginoux

Lines 29-31: "*Many of the traditional dust emission models (TEM) assume that the Earth's land surface is devoid of vegetation, then adjust the dust emission using a vegetation cover complement, and finally calibrate the magnitude of simulated emissions to dust in the atmosphere*"
The calibration is mostly related to numerical discretization of the momentum and continuity equations. Emission of dust in numerical models depends on the discretization of surface winds. The surface winds are inferred from the pressure level wind vectors derived by solving numerically the momentum equations. The numerical discretization of these equations will be affected by the numerical resolution. Obviously higher resolution will resolve sharp topographic variations with stronger downslope winds. On the other hand, flat terrain without roughness elements using low resolution will generate stronger gustiness. So, changing model resolution has a non-trivial effect on surface winds. Concerning dust emission, the flux depends on the cubic power of surface winds (see Equation 2), which will amplify wind bias related to model resolution. This implies that "tuning" dust emission is a required method to simulate scale-aware tracer with numerical model. This is also true for other tracers, such as sea salt emission from the oceans.

Lines 82-83: "*The common approach to modeling dust emission in ESMs uses globally constant values of aerodynamic roughness length ($z_0$), which are static over time and fixes $R(z_0) \approx 0.91$.*"
This is incorrect. In ESMs the momentum roughness length is calculated at every time steps and in every grid cells as a function of terrain variations, vegetation cover, snow cover, etc.

Line 81-85: "*The common approach to modelling dust emission in ESMs… This emission is then reduced by a function of vegetation cover and ultimately 'tuned' down to match observed in the atmosphere.*"
I am unaware of any ESMs who have implemented dust emission as described. I can certainly speak for NASA and GFDL models (Ginoux et al., 2001; Evans et al., 2016).

Lines 103-104: "*The $u_{s*}$ is obtained directly from $\omega_{ns}$, the normalised and rescaled shadow (1-albedo), enabling an albedo-based dust emission model (AEM; see Appendix for full description of the implementation)*"
Do I read correctly that you are scaling the friction velocity using 3 parameters with an exponential function of $\omega_{ns}$? Am I right that you will have to rescale $\omega_{ns}$ for any other satellite instruments with different viewing angles or radiometric characteristics? Is this not a global tuning?

Line 128-132: "*Evans et al., 2016*"
The characterization of GFDL model (Evans et al. 2016) is not correct. You may want to read the paper. We are not using E=1-Av. The bare surface is calculated using an exponential function of the canopy (LAI) and stems, twigs, litters (SAI). The dust emission is calculated in each land tiles (primary, secondary vegetation, pasture and cropland) independently. Then the flux of dust is passing through a flux-exchanger into the atmosphere while a flux down from turbulence and settling is going in the land model. The latest ESM4 includes also tiles from fires and rangeland, in addition for taking into account slopes (Dunne et al., 2020; Horowitz et al., 2020). I will disagree with you when calling such detailed and consistent modeling of dust cycle a "crude model representation"

Lines 133-135: "*When the TEMs are applied in dust-climate ESMs it is assumed that this parameterization is adequate for climate projections. In contrast, the albedo-based scheme for sediment flux and dust emission (AEM; Eqs. 3, 4 & 5) represents the drag partition physics without pre-tuning to a fixed land surface condition, without the need for E, and thereby removes these additional sources of uncertainty.*"
The main point of using climate model has been missed here. Despite their approximations, ESMs simulate the different Earth's climate systems consistently over time using different projection scenarios. While the proposed used of $\omega_{ns}$ (the normalized and rescaled shadow) is considered fixed (beyond MODIS period), ignoring vegetation and land use changes. The AEM technique is inadequate for future or past climates.

Lines 145-149: "*To understand the extent to which AOD estimates the spatial variation in dust emission magnitude and frequency we calculated the probability of dust occurrence modeled by the dust optical depth (DOD>0.2) using the criteria established previously (Ginoux et al., 2012). We note the stated limitations of DOD to be largely restricted to bright land surfaces in the visible wavebands which implies reduced performance over areas where vegetation is present.*"
This sentence contains several misunderstandings of our latest method developed with my co-authors to derive DOD.

In our 2012, we used the collection 5.1 of MODIS Deep Blue (DB), which provided aerosol products over bright surfaces. Since 2013 Collection 6 MODIS DB aerosol products have been extend to cover most (without snow or cloud cover) land surfaces (Sayer et al., 2014). All subsequent papers deriving DOD is using Collection 6.1 MODIS DB (e.g. Pu and Ginoux, 2017, 2018a, 2018b, 2020; Yu and Ginoux, 2021). A second update is the method to calculate DOD. Since Pu and Ginoux (2017), DOD is calculated using a quadratic function of aerosol optical depth (AOD) and the single scattering albedo (SSA). In our 2012 paper, DOD is calculated using an on/off switch depending on the value of the Angstrom Exponent (AE). Then a threshold is applied to detect the highest frequency to correspond to actual dust sources. The method has been compared to independent geomorphological data over the Chihuahuan desert (Baddock et al., 2016) to prove that MODIS DB DOD is able to successfully detect high-resolution dust sources. It will be necessary to add a note the text stating that you are referring to an old dataset long replaced by thoroughly validated values using the latest MODIS aerosol products. Preferably, you replace the sentence by referring to more recent thoroughly validated values using the latest MODIS aerosol products.

Line 149-150: "*To calculate DOD, we used wavebands available from monthly Moderate Resolution Imaging Spectroradiometer (MODIS; MOD08 M3 V6.1) at a 1-degree pixel resolution (Platnick, 2015)*"
These are monthly gridded products at 1 x1 degree resolution from 1 km daily pixels. Using coarse resolution monthly aerosol products is totally inadequate to compare with point source dust plumes. The difference with MODIS DB DOD from my team is that it is 0.1 x 0.1 degree twice-daily products, which was shown to be appropriate to detect tiny dust sources (Baddock et al., 2016).
It is unclear how you get these DOD. Citing Platnick, 2015 is not helpful. What MODIS Level 3 products are you using? Are you using Dark Target or Deep Blue algorithm? Maybe you are using a blend of the two algorithms? DOD is not part of these products, so a description on how you obtain DOD from MODIS would be helpful, especially that you assimilate it to Ginoux et al. (2012) which is outdated.

Line 154: "*We also provided a theoretical basis for TEMs formulation to be incorrect.*"
I commented earlier that your description of TEMs formulation is mostly incorrect.

Line 166: "*using MODIS data at 250 m spatial resolution with visible to thermal infrared wavebands*"

This is incorrect. Only bands 1 and 2 are provided at 250 m. Bands 3 to 7 are at 500 m resolution. Bands 8 to 36 are at 1 km resolution. Red is band 2, blue is band 3 and green band 4. Deep blue is band 8 or 1 km pixel. Then 10 x 10 pixels are aggregated to provide 10x10 km Level 2 daily aerosol products.

Line 171: "*DOD modelled frequency describes DOD > 0.2*"
In Baddock et al. (2016), we used DOD>0.75 over the Chihuahuan desert, but Pu et al. (2020) used 2 threshold values (0.2 and 0.02) depending on the continent. Choosing a DOD threshold should be adapted to the objectives of the study but using gridded 1-degree monthly products is too coarse spatially and temporally to study dust sources.

Lines 248-249: "*the TEM is driven by wind speed attenuated by aerodynamic roughness which is fixed over space and static over time,…*"
If the surface conditions don't change (no snow, no vegetation or land use changes) this will be true, but most ESMs (or TEMs) include such changes when resolving the boundary layer properties.

References

Baddock, M. C., Ginoux, P., Bullard, J. E., and Gill, T. E.: Do MODIS-defined dust sources have a geomorphological signature?, Geophys. Res. Lett., 43, 2606–2613, https://doi.org/10.1002/2015GL067327, 2016.

Dunne, J. P., Horowitz, L. W., Adcroft, A. J., Ginoux, P., Held, I. M., John, J. G., et al.: The GFDL Earth System Model Version 4.1 (GFDL-ESM 4.1): Overall coupled model description and simulation characteristics. *Journal of Advances in Modeling Earth Systems*, 12, e2019MS002015. https://doi.org/10.1029/2019MS002015, 2020.

Evans, S., Ginoux, P., Malyshev, S., and Shevliakova, E.: Climate-vegetation interaction and amplification of Australian dust variability, Geophys. Res. Lett., 43, 11823–11830, https://doi.org/10.1002/2016gl071016, 2016.

Ginoux, P., M. Chin, I. Tegen, J. M. Prospero, B. Holben, O. Dubovik, and S.-J. Lin: Sources and distributions of dust aerosols simulated with the GOCART model, *J. Geophys. Res.*, 106, 20,255– 20,274, 2001.

Ginoux, P., Prospero, J. M., Gill, T. E., Hsu, N. C., and Zhao, M.: Global-scale attribution of anthropogenic and natural dust sources and their emission rates based on MODIS Deep Blue aerosol products, Rev. Geophys., 50, RG3005, https://doi.org/10.1029/2012RG000388, 2012.

Horowitz, L. W., Naik, V., Paulot, F., Ginoux, P. A., Dunne, J. P., & Mao, J., et al.: The GFDL global atmospheric chemistry-climate model AM4.1: Model description and simulation characteristics. *Journal of Advances in Modeling Earth Systems*, 12, e2019MS002032. https://doi.org/10.1029/2019MS002032, 2020

Platnick, S.: MODIS Atmosphere L3 Monthly Product, NASA MODIS Adaptive Processing System, Goddard Space Flight Center, 10.5067/MODIS/MOD08_M3.006, 2015.G

Pu, B. and Ginoux, P.: Projection of American dustiness in the late 21st century due to climate change, Sci. Rep., 7, 5553, https://doi.org/10.1038/s41598-017-05431-9, 2017.

Pu, B. and Ginoux, P.: Climatic factors contributing to long-term variations in surface fine dust concentration in the United States, Atmos. Chem. Phys., 18, 4201–4215, https://doi.org/10.5194/acp-18-4201-2018, 2018a.

Pu, B. and Ginoux, P.: How reliable are CMIP5 models in simulating dust optical depth?, Atmos. Chem. Phys., 18, 12491–12510, https://doi.org/10.5194/acp-18-12491-2018, 2018b.

Pu, B., Ginoux, P., Guo, H., Hsu, N. C., Kimball, J., Marticorena, B., Malyshev, S., Naik, V., O'Neill, N. T., Pérez García-Pando, C., Paireau, J., Prospero, J. M., Shevliakova, E., and Zhao, M.: Retrieving the global distribution of the threshold of wind erosion from satellite data and implementing it into the Geophysical Fluid Dynamics Laboratory land–atmosphere model (GFDL AM4.0/LM4.0), Atmos. Chem. Phys., 20, 55–81, https://doi.org/10.5194/acp-20-55-2020, 2020.

Sayer, A. M., Munchak, L. A., Hsu, N. C., Levy, R. C., Bettenhausen, C., and Jeong, M.-J. (2014), MODIS Collection 6 aerosol products: Comparison between Aqua's e-Deep Blue, Dark Target, and "merged" data sets, and usage recommendations, *J. Geophys. Res. Atmos.*, 119, 13,965– 13,989, doi:10.1002/2014JD022453.

Yu, Y. and Ginoux, P.: Assessing the contribution of the ENSO and MJO to Australian dust activity based on satellite- and ground-based observations, Atmos. Chem. Phys., 21, 8511–8530, https://doi.org/10.5194/acp-21-8511-2021, 2021.

---

## Author Comment (AC3)

**Dear Reviewer,**

**Many thanks for taking the time to provide comments on our manuscript. To each of your comments / queries, we provide responses below using indented bullet points with a bold blue typeface.**

**General Comments**

This study introduced a novel method based on dynamic land cover change (albedo or earth's shadow) to quantify dust emission with grid precision and thus overcomes the biases from the traditional approach that estimated dust emissions based on constant spatial vegetation distribution from bare soil assumptions. The aim is to investigate point source emission detected by satellites observation varying with time and space. They found that both approaches/models overestimated the occurrence of dusty days, which is mainly from soil wind friction velocity. The more the model overestimates the soil wind friction velocity, the more it entrains high sediment flux once the threshold is exceeded. Therefore, the albedo-based model generates lower emissions than the traditional model due to the new formulation of soil wind friction velocity obtained as function albedo, roughness and horizontal wind velocity. This newly developed albedo-based model suggested to mimic the soil bareness and vegetation cover before and after dust emissions, and the results proved moderately good performance. This study is important and has potential impact for modelling community especially in quantifying effective emission of dust. Some questions for the experiment design and results are need to be addressed prior to the publication.

**Specific Comments**

1. Dust Point Source locations are only shown on a small-domain map in Figure 1; Later, the authors described the roughness, wind speed, and dust flux in Figure 5 over a larger-domain map, would the authors show the DPS over a larger-domain map, such as the continental US (CONUS)? And, if possible, a map of North America is preferred to display district boundaries, deserts, vegetation and etc.

- **The dust emission point source (DPS) data are from all the existing studies for which data are available. All the DPS data in North America used in this evaluation are shown in Figure 1 of the manuscript. The spatial extent of those DPS data is in, and around, New Mexico. No other DPS data are currently available.**
- **To make the previous point clear, we show these DPS data inset over the larger North American domain (Figure R1). This new figure will be included in the revised manuscript.**

[Figure]

- **Figure R1. Location and publication source (Kandakji et al., 2020; Lee et al., 2012; Baddock et al., 2011) inventory in New Mexico and Texas between 2001-2016 (Kandakji), 2001-2009 (Lee) and in 2001-2009 in the Chihuahuan Desert and New Mexico (Baddock) using satellite observed dust emission point sources (DPS) set against a background of total wind friction velocity ($u_*/U_{10}$) derived from MODIS albedo (500 m). The inset shows the location of DPS data in North America.**

    AOD represents the total aerosol burden in the atmosphere. DOD is meant for detecting dust particles in the atmosphere. In this study, the authors preferred to employ a threshold of 0.2 for DOD (DOD>0.2) as from previous study of Ginoux et al (2012) to separate dust from background over North America during spring season.

- **We followed the methodology using the established criteria and using a threshold of 0.2 for dust optical depth (DOD>0.2; Ginoux et al., 2012) to avoid misrepresenting the DOD.**

    However, Ginoux et al (2012) used the threshold DOD>0.25 for most of the regions, and the threshold from this previous study is retrieved from the MODIS-DB L2 product at the 10 km x10 km grid resolution that is much finer than 1 degree resolution used in this study.

- **Ginoux et al. (2012 bottom of page 10) state that in North America, the highest frequency of dust events is found in the southwestern U.S. and northern Mexico. Along the border between the U.S. and northern Mexico, events with DOD>0.2**

**appear as frequently as 30% of the time in MAM. This is in agreement with the long-term record of visibility data at El Paso (Texas).**

- **Satellite observed dust emission point sources (DPS) have an uncertainty of around +/-2 km (Kandakji et al., 2020) due to the phase difference between timing of dust emission and availability of the imagery. At the point scale, unexplained (nugget) variance between DPS is reduced by aggregating data over time to months, and over space to 1 degree grid boxes. This approach is well-established in geostatistical literature and used across multiple disciplines and tackles the issue of incompatible scales described by Gotway and Young (2002).**
- **We used monthly MODIS Deep Blue Collection 6 data (MOD08 M3 V6.1) to establish DOD across 1-degree grid boxes compatible with the DPS data.**

  Given that the authors in this study overestimated the frequency (Figure 3) even with the lower DOD threshold of 0.2 without quality control, why didn't the authors sample DOD with quality flag at higher resolution and then average over one degree resolution? Otherwise, is 0.2/0.25 reasonable for one-degree resolution? Over such a large grid, smaller value may be preferred? please clarify.

- **Consistent with previous studies using DOD to display spatial variability, we show in Figure 3 of the manuscript, the discrepancy between the spatial variation in dust emission point sources (DPS; Figure 3a) and dust in the atmosphere measured using DOD (Figure 3b).**
- **To investigate the reviewer's question about the sensitivity of the maps to the threshold used, we reproduced Figure 3b in the manuscript with different thresholds: dust optical depth (a) DOD>0, (b) DOD>0.1, (c) DOD>0.2, and (d) DOD>0.25.**
- **These results (Figure R2) show that there is very little difference in P(DOD>threshold) along the USA and Mexico border, and small differences in areas exceeding threshold more frequently at smaller thresholds further north and east.**
- **These new results confirm that the choice of threshold around DOD>(0.2-0.25) makes little difference to our results and interpretations.**
- **Notably, where no threshold is applied (Fig. R2a), dust occurrence increases in the northern areas of the figure. This type of threshold is not applied in the DOD literature because it is vulnerable to erroneous observations due to atmospheric and surface conditions that would otherwise be screened out with the application of a threshold.  We will include these findings in the Appendix of the revised manuscript.**

[Figure]

**Figure R2. Comparison between the probability of MODIS dust optical depth (DOD>T) where T=0 (a), T=0.1 (b), T=0.2 (c) and T=0.25 (d) during the study period 2001-2016. All available MODIS DOD data were used, quality flags were not used to filter these data. The missing value of the pixel in the south-east of MODIS DOD is evident in the original data and has not been removed during processing.**

2.  The results showed that high dust emissions were generated mainly from the Great Plains extending from Montana, Wyoming, Dakota, Colorado, New Mexico, and Texas, and slight dust emission were from the semi-arid and arid regions of the western deserts (Sonoran, Chihuahua, Mohave and great basin deserts). Therefore, I

think the authors should also explain why those semi-arid and arid regions did not have any DPS.

- **The reviewer notes that dust emission (evident in Figure 5 of the manuscript) occurs in regions other than those where dust emission point source data (DPS) has previously been measured. The reason we do not show any DPS data for those regions is that we did not use any DPS data in those regions. This is because there are no available DPS data outside of those regions we showed.**
- **The identification of DPS data is a highly time-consuming and labour-intensive activity. Consequently, there are few (published) studies relative to the large number of dust source regions. We will use this last sentence in the revised manuscript to explain the availability of DPS data.**

3. How does the study incorporate the soil texture/ soil type especially particle size threshold for starting the dust saltation? More explanation is preferred.

- **There are several places in the existing manuscript where soil texture is used in the sediment transport *Q* and dust emission *F* modelling. Firstly, equations 1-3 in the Introduction (of the main text) demonstrates $Q(d)$ where in the entrainment threshold u*ts(d). Secondly, equation 5 in the Introduction demonstrates that dust in the atmosphere is a function of the relative particle size surface area and of the clay content.**
- **The precise description of how soil texture is included in the modelling is provided in the Appendix and includes a description of the entrainment threshold and the soil moisture function, which adjusts u*ts, depending on the clay fraction. In the revised manuscript we will ensure that the Introduction refers more to the Appendix.**

**Minor Comments**

4. The paragraph from 231-237 describes Figure 2a for the albedo-based model. It seems that the results from the smooth and rough cases overlap. Please clarify if they are identical.

- **In the manuscript, L231-237 describes the model assumptions applicable to a traditional model and the albedo-based model (AEM).**
- **Figure 2a is described from L. 297. The AEM dust emission with a smooth condition (us*/Uh=0.035) represented by a dotted line which extends to the entrainment threshold of us*=0.2. The AEM dust emission under a rough condition (us*/Uh=0.022) is represented by a solid black line. The AEM dust emission response overlaps.**

- **In the revised manuscript we will change the line styles to make the overlap visible.**

5.  In Figure 5 and 6, $U_h$ should be replaced by $U_{10}$. Please also correct others if any.

- **Thanks for identifying those typographic mistakes. In the revised manuscript, we will rectify those errors and any others that we find.**